# Modified Blind Equalization Algorithm Based on Cyclostationarity for Contaminated Reference Signal in Airborne PBR

**DOI:** 10.3390/s20030788

**Published:** 2020-01-31

**Authors:** Shuai Guo, Jun Wang, Hui Ma, Jipeng Wang

**Affiliations:** National Laboratory of Radar Signal Processing, Xidian University, Xi’an 710071, China; xdguoshuai@163.com (S.G.); h.ma@xidian.edu.cn (H.M.); jpwang_2@stu.xidian.edu.cn (J.W.)

**Keywords:** airborne passive bistatic radar, multipath signal, modified blind equalization algorithm, cyclostationarity, complex value BP neural network

## Abstract

In airborne passive bistatic radar (PBR), the reference channel toward the opportunity illuminator is applied to receive the direct-path signal as the reference signal. In the actual scenario, the reference signal is contaminated by the multipath signals easily. Unlike the multipath signal in traditional ground PBR system, the multipath signal in the airborne PBR owns not only the time delay but also the Doppler frequency. The contaminated reference signal can cause the spatial-temporal clutter spectrum to expand and the false targets to appear. The performance of target detection is impacted severely. However, the existing blind equalization algorithm is unavailable for the contaminated reference signal in airborne PBR. In this paper, the modified blind equalization algorithm is proposed to suppress the needless multipath signal and restore the pure reference signal. Aiming at the Doppler frequency of multipath signal, the high-order moment information and the cyclostationarity of source signal are exploited to construct the new cost function for the phase constraint, and the complex value back propagation (BP) neural network is exploited to solve the constraint optimization problem for the better convergence. In final, the simulation experiments are conducted to prove the feasibility and superiority of proposed algorithm.

## 1. Introduction

Without the purposely-built transmitter, the passive bistatic radar (PBR) can be described as the “receiver-only” radar. Relaying on the unique advantages, the PBR has been attracting the international research interest recently [1]. The traditional PBR, which is equipped to the ground platform, exploits the civilian or commercial signal as the opportunity illuminator signal to detect the potential target, such as frequency modulate (FM) signal [2,3], global system for mobile communication (GSM) signal, digital audio broadcast (DAB) signal, 4G long-term evolution (LTE) signal [4], digital video broadcast-terrestrial (DVB-T) signal [5,6], active radar signal [7], and so on. Currently, the theoretical study on the PBR system develops quickly, and a series of studies on the interference and clutter suppression make the further progress in [8,9,10].

The PBR detection, as the important complement approach of the active radar detection, has the great potentiality to be developed. One of the important developing directions is the airborne PBR that owns the more flexible maneuverability and the greater detection power [11,12]. The advantage of electromagnetic silence brings more security to the airborne early warning (AEW) aircraft. The detection technology of airborne PBR attracts the wide attention and develops rapidly within a very short time.

However, the airborne PBR is also faced with the severe clutter case as same as the airborne active radar. Because of the moving platform, the received clutter has the property of space-time coupling. The Doppler spectrum of clutter expands seriously and the target echo is obscured by the spatial-temporal clutter. Aiming at the clutter issue, the displaced phase center antenna (DPCA) is used to suppress the spatial-temporal clutter by compensating the platform movement [13,14]. Then, the space-time adaptive processing (STAP) is proposed to separate the target echo and the expanded clutter from a joint Spatial-Doppler dimension in airborne PBR [15,16]. Because of the large number of space-time degree of freedoms (DoFs), the real-time processing of full STAP brings the huge calculating pressure to the radar system. Hence, the reduced-rank and partial STAP are developed to achieve the approximately optimal performance. The existing reduced-rank STAP algorithms including the AEP-STAP, JDL-STAP, and 3DT-STAP are widely applied to the practical project. Meanwhile, the full STAP needs a large number of training snapshots that are independent and identically distributed (IID). The ideal condition is difficult to satisfy in the actual scenario. The STAP in nonhomogeneous environment is researched in [17,18]. The knowledge-aided STAP and sparsity-based STAP are developed to enhance the detection performance in the nonhomogeneous environment [16,17,18]. The knowledge-aided STAP utilizes the priori information to improve the estimation performance of clutter covariance matrix in [19]. It is assumed that the clutter is sparse in the spatial-temporal domain. The sparsity-based STAP exploits the weight vector and the over-complete space-time steering dictionary to represent the clutter subspace approximately in [20,21]. The sparsity-based STAP provides the high-resolution of scene and exhibits the better performance with the less training snapshot. 

The traditional PBR system exploits the reference channel toward the opportunity illuminator to receive the direct-path signal as the reference signal for the subsequent matching processing. In the actual scenario, the received reference signal is also contaminated by the multipath signals easily. In the ground PBR system, the multipath signal can be treated as the delayed direct-path signal. Aiming at the contaminated reference signal, some equalization algorithms based on constant modulus algorithm (CMA) are proposed to suppress the multipath signal in the reference channel [22,23]. The fractional order CMA is proposed to equalize the propagation channel in [22]. The authors of [23] propose the space-time CMA for the multipath removal in the reference channel. The algorithm joins the space domain and time domain to restrain the multipath signal, but the algorithm which requires the array antenna to collect the reference signal brings more complexity to the PBR system.

The airborne PBR is also faced with the unique issue that the reference signal is contaminated by the multipath signal. With the impact of multipath propagation channel, the reference signal will be blurred by the multipath signal. The contaminated reference signal will degrade the detection performance of existing STAP algorithm. The reason is that the multipath signal matched with the spatial-temporal clutter can expand the clutter power spectrum, and the multipath signal matched with the target echo can cause the false target which increases the false alarm probability. Additionally, the multipath signal in the airborne PBR is different from the one in the ground PBR. The multipath signal in the ground PBR can be treated as the time-delay replica of direct-path signal without the Doppler frequency, but the multipath signal in airborne PBR has not only the time delay but also the Doppler frequency. Thus, the traditional multipath suppression algorithms mentioned above are not applicable in the airborne PBR system. Aiming at the special case of contaminated reference signal in the airborne PBR, the expanded clutter suppression algorithm based on sparse representation is proposed in [24]. The algorithm constructs the cost function with a sparse constraint, and the relation of spatial-temporal clutters which are matched with the direct-path signal and the multipath signal is derived. The expanded clutter matched with multipath signal will be eliminated by iteratively deriving weight vector. However, the false target caused by the multipath signal can be detected in the range dimension inevitably. The algorithm in [24] ignores the issue of false target.

In this paper, the modified blind equalization algorithm is proposed to process the contaminated reference signal directly. The proposed algorithm modifies CMA to restrain the multipath signal and restore the pure reference signal. The cyclostationarity, which is applied to the blind signal processing widely, can be used to modify the traditional blind equalization algorithm in this paper. The source signal of interest can be extracted by the unique cyclostationarity frequency of source signal in [25,26]. Aiming at the multipath signal with Doppler frequency, the high-order moment information and cyclostationarity of source signal are exploited to construct the new cost function in order to restore the phase of source signal. The complex value back propagation (BP) neural network is exploited to equalize the contaminated reference signal for the better convergence. Finally, the proposed algorithm combines STAP to suppress the spatial-temporal clutter and achieve the target detection.

The structure of this paper is organized as follows. Section 2 introduces the considerable issue caused by the multipath signal in the reference channel, and the impact on the detection performance is derived. The modified blind equalization algorithm is proposed in Section 3, and this section describes the derivation process of proposed algorithm in detail. Section 4 conducts the simulation experiments to prove the feasibility of proposed algorithm in this paper, and the performance of proposed algorithm is analyzed. Finally, the conclusion is drawn in Section 5.

## 2. Signal Model and Case-Study Scenario

The airborne PBR system utilizes the reference channel and the echo channel to receive and process the reference signal and the echoes respectively. The echo channel collects the low-level target echoes with the array antenna toward the surveillance area, and the reference channel receives the direct-path signal as the reference signal of matching processing. In the complex environment as shown by Figure 1, the reference signal is contaminated by the multipath signals that are reflected by the hills or buildings. So the received reference signal includes the desired direct-path signals and the undesired multipath signals. The contaminated reference signal can cause the expanded spatial-temporal clutter spectrum and the false targets. The impact of contaminated reference signal is derived in this section.

First, the model of contaminated reference signal is introduced. In airborne PBR system, the direct-path signal can be treated as the delayed replica of transmitted signal with the Doppler frequency. The complex envelope of direct-path signal can be modeled as: (1)$$S_{d}\left(t\right)=A_{d}\sum_{m=0}^{M-1}u\left(t-\tau_{d}-mT_{r}\right)e^{j2\pi f_{d}t}$$
where, $A_{d}$ is the complex amplitude of direct-path signal, $\tau_{d}$ is the time delay, and $f_{d}$ is the Doppler frequency caused by the airborne platform. $\sum_{m=0}^{M-1}u\left(t-mT_{r}\right)$ is the complex envelope of transmitted signal, M is the pulse number in a coherent process interval (CPI), and $T_{r}$ is the pulse repeat interval.

Generally, the multipath signal can be treated as the weakening and time-delayed direct-path signal with the different Doppler frequency. The complex envelope of multipath signal can be modeled as:(2)$$S_{mul}\left(t\right)=h\left(t\right)*S_{d}\left(t\right)=\sum_{p=1}^{N_{p}}A_{p}\sum_{m=0}^{M-1}u\left(t-\tau_{p}-mT_{r}\right)e^{j2\pi f_{p}t}$$
where, (∗) denotes the convolution operation, h(t) is the impulse response of multipath propagation channel. $A_{p}$ is the amplitude of the *p*th impulse response, $\tau_{p}$ is the time delay of the *p*th multipath signal, and $N_{p}$ is the number of multipath signals. 

The reference signal contaminated by the multipath signal is modeled as:(3)$$\begin{array}{cc} S_{ref}\left(t\right)= & S_{d}\left(t\right)+S_{mul}\left(t\right)+n\left(t\right)\\ & =A_{d}\sum_{m=0}^{M-1}u\left(t-\tau_{d}-mT_{r}\right)e^{j2\pi f_{d}t}+\sum_{p=1}^{N_{p}}A_{p}\sum_{m=0}^{M-1}u\left(t-\tau_{p}-mT_{r}\right)e^{j2\pi f_{p}t}+n\left(t\right)\\ & =A_{d}s_{o}\left(t-\tau_{d}\right)e^{j2\pi f_{d}t}+\sum_{p=1}^{N_{p}}A_{p}s_{o}\left(t-\tau_{p}\right)e^{j2\pi f_{p}t}+n_{ref}\left(t\right) \end{array}$$
where, $n_{ref}\left(t\right)$ denotes the reference channel noise.

Then, the impact of multipath signal on the spatial-temporal clutter spectrum is derived. The space-time character of clutter is described as follows. The spatial and temporal steering frequency is shown as:(4)$$\begin{array}{l} \omega_{s}\left(\theta ,\varphi \right)=\frac{2\pi d}{\lambda }\cos\theta \cos\varphi \\ \omega_{t}\left(\theta ,\varphi \right)=\frac{2\pi V}{\lambda f_{r}}\cos\theta \cos\varphi \end{array}$$
where, θ and φ are the pitching angle and azimuth angle, d and λ are the array element interval and the wavelength. V and $f_{r}$ are the platform velocity and the pulse repeat frequency (PRF).

The spatial-temporal clutter received by the *n*th antenna element form the *l*th clutter range bin is modeled as:(5)$$S_{n,l}\left(t\right)=\sum_{i=1}^{N_{c}}A_{i}\sum_{m=0}^{M-1}u\left(t-\tau_{l}-mT_{r}\right)e^{j2\pi f_{i}t+j2\pi n\vartheta_{n,i}}$$
where, $A_{i}$ is the complex amplitude, $\tau_{l}$ is the delay time of the *l*th clutter range bin, $N_{c}$ is the number of spatial-temporal clutter, $f_{i}$ is the Doppler frequency of the *i*th clutter, and $\vartheta_{n,i}$ is the angle of arrival of the *i*th clutter.

The matched output of the contaminated reference signal and the spatial-temporal clutter is shown as:(6)$$\begin{array}{cc} \chi_{n,l}\left(t\right) & =\int S_{n,l}\left(\xi \right)S_{ref}^{*}\left(\xi -t\right)d\xi \\ & =\sum_{i=1}^{N_{c}}\varepsilon_{i}e^{j2\pi n\vartheta_{i}}\sum_{m=0}^{M-1}e^{j2\pi mT_{r}\left(f_{i}-f_{d}\right)}r_{m}\left(t-\left(\tau_{i}-\tau_{d}\right)-mT_{r}\right)\\ & +\sum_{p=1}^{N_{p}}\sum_{i=1}^{N_{c}}\varepsilon_{i,p}e^{j2\pi n\vartheta_{i}}\sum_{m=0}^{M-1}e^{j2\pi mT_{r}\left(f_{i}-f_{p}\right)}r_{m}\left(t-\left(\tau_{i}-\tau_{p}\right)-mT_{r}\right)+\chi_{noise}\left(t\right)\\ & ={\chi^{\prime }}_{n,l}\left(t\right)+{\chi^{\prime \prime }}_{n,l}\left(t\right)+\chi_{noise}\left(t\right) \end{array}$$
where, ${\chi^{\prime }}_{n,l}\left(t\right)$ is the matched output of the direct-path signal and the spatial-temporal clutter, ${\chi^{\prime \prime }}_{n,l}\left(t\right)$, which is the second part of Equation (6), is the additional matched output of the multipath signal and the spatial-temporal clutter, and $\chi_{noise}\left(t\right)$ is the matched output about the channel noise. The multipath signal in the reference signal causes the spatial-temporal clutter spectrum to expand.

Finally, the reason of false target is derived. The target echo is modeled as:(7)$$S_{tar}\left(t\right)=\sum_{k=1}^{N_{t}}A_{k}\sum_{m=0}^{M-1}u\left(t-\tau_{k}-mT_{r}\right)e^{j2\pi f_{k}t}$$
where, $A_{k}$ is the complex amplitude of the *k*th target, $\tau_{k}$ and $f_{k}$ are the delay time and Doppler frequency respectively.

The matched output of the contaminated reference signal and the target echo is shown as:(8)$$\begin{array}{cc} \chi_{tar}\left(t\right) & =\int S_{tar}\left(\xi \right)S_{ref}^{*}\left(\xi -t\right)d\xi \\ & =\sum_{k=1}^{N_{t}}\varepsilon_{k}\sum_{m=0}^{M-1}e^{j2\pi mT_{r}\left(f_{k}-f_{d}\right)}r_{m}\left(t-\left(\tau_{k}-\tau_{d}\right)-mT_{r}\right)\\ & +\sum_{p=1}^{N_{p}}\sum_{k=1}^{N_{t}}\varepsilon_{k,p}\sum_{m=0}^{M-1}e^{j2\pi mT_{r}\left(f_{k}-f_{p}\right)}r_{m}\left(t-\left(\tau_{k}-\tau_{p}\right)-mT_{r}\right)+\chi_{noise}\left(t\right)\\ & =\sum_{k=1}^{N_{t}}S_{k}^{R}\left(t\right)+\sum_{p=1}^{N_{p}}\sum_{k}^{N_{t}}S_{k,p}^{F}\left(t\right)+\chi_{noise}\left(t\right) \end{array}$$
where, the *k*th real target is detected in $\tau_{k}-\tau_{d}$ range bin. The false targets, which are derived from the *k*th real target by the *p*th multipath, are detected in $\tau_{k}-\tau_{p}$ range bin. The false target degrades the detection performance of airborne PBR system.

In conclusion, the reference signal contaminated by the multipath signal expands the spatial-temporal clutter spectrum and causes the false targets in the nearby range bins. The impact of contaminated reference signal cannot be ignored in the airborne PBR. The feasible algorithm should be proposed to solve the issue.

## 3. Proposed Blind Equalization Algorithm

In order to overcome the issue of contaminated reference signal by the multipath signal, the modified blind equalization algorithm is proposed in this paper. In the equalization processing, the Doppler frequency of multipath signal should be considered. In this paper, the proposed algorithm exploits the high-order moment information and the cyclostationarity frequency of source signal to construct the new cost function. The constraint optimization problem can be solved by the complex value BP neural network for the better convergence.

### 3.1. Cyclostationarity

The cyclostationarity, which owns the better signal selectivity, can be exploited to extract the interesting signals. The second-order cyclic conjugate autocorrelation is shown as:(9)$$R_{ss}^{\beta }\left(\tau \right)=E_{t}\left\{R_{ss}\left(t,\tau \right)e^{-j2\pi \beta t}\right\}$$
where, $E_{t}\left\{\cdot \right\}$ denotes the processing of mathematical expectation, and $R_{ss}\left(t,\tau \right)=E_{t}\left\{s\left(t\right)s\left(t+\tau \right)\right\}$ denotes the conjugate autocorrelation of source signal s(t). The signal selectivity of cyclic conjugate autocorrelation is shown by:(10)$$R_{pq}^{\beta }\left(\tau \right)=\left\{ \begin{array}{l} E_{t}\left\{s_{p}\left(t\right)s_{q}\left(t+\tau \right)e^{-j2\pi \alpha_{p}t}\right\}>0,p=q\\ E_{t}\left\{s_{p}\left(t\right)s_{q}\left(t+\tau \right)e^{-j2\pi \alpha_{p}t}\right\}=0,p\neq q\\ E_{t}\left\{s_{p}\left(t\right)s_{q}\left(t+\tau \right)e^{-j2\pi \alpha_{p}t}\right\}=0,\alpha_{p}\neq \alpha_{q} \end{array}\right.$$

The cyclostationarity of contaminated reference signal is analyzed, and the cyclostationarity frequencies of direct-path signal and multipath signal can be determined. The cyclic conjugate autocorrelation of contaminated reference signal is shown as
(11)$$\begin{array}{cc} R_{ss}^{\beta }\left(0\right) & =E_{t}\left\{S_{ref}\left(t\right)S_{ref}\left(t\right)e^{-j2\pi \beta t}\right\}\\ & =E_{t}\left\{A_{d}^{2}s_{0}^{2}\left(t-\tau_{d}\right)e^{j2\pi 2f_{d}t}e^{-j2\pi \beta t}\right\}+E_{t}\left\{2\sum_{p=1}^{N_{p}}A_{d}A_{p}s_{0}\left(t-\tau_{d}\right)s_{0}\left(t-\tau_{p}\right)e^{j2\pi \left(f_{p}+f_{d}\right)t}e^{-j2\pi \beta t}\right\}\\ & +E_{t}\left\{\sum_{q=1}^{N_{p}}\sum_{p=1}^{N_{p}}A_{q}A_{p}s_{0}\left(t-\tau_{q}\right)s_{0}\left(t-\tau_{p}\right)e^{j2\pi \left(f_{p}+f_{q}\right)t}e^{-j2\pi \beta t}\right\}+R_{\mathrm{noise}}\\ & =A_{d}^{2}R_{dd}^{\beta -2f_{d}}\left(0\right)+2\sum_{p=1}^{N_{p}}A_{d}A_{p}R_{dp}^{\beta -\left(f_{p}+f_{d}\right)}\left(\tau_{d}-\tau_{p}\right)+\sum_{q=1}^{N_{p}}\sum_{p=1}^{N_{p}}A_{q}A_{p}R_{qp}^{\beta -\left(f_{q}+f_{p}\right)}\left(\tau_{q}-\tau_{p}\right)+R_{\mathrm{noise}} \end{array}$$
where, the first part of Equation (11) is the cyclic conjugate autocorrelation of direct-path signal, the second part is the cyclic cross-correlation of direct-path signal and multipath signal, and the third part is the cyclic autocorrelation of multipath signal. The $R_{\mathrm{noise}}$ is the sum of cyclic cross-correlations about the channel noise. According to Equation (11), the peaks of cyclic autocorrelation appear at $\beta =2f_{d},f_{d}+f_{p},f_{p}+f_{q}\left(p,q=1,\ldots N_{p}\right)$. With the above analysis, the cyclostationarity can be applied into the derivation of proposed algorithm. 

### 3.2. Modified Blind Equalization Algorithm Based on Cyclostationarity and BP Network

The traditional CMA, which is one of the well-known blind equalization algorithms, exploits the finite impulse response (FIR) filter to equalize the contaminated signal, and the output of filter is shown as:(12)$$\hat{y}\left(t\right)=w^{H}\left(t\right)x\left(t\right)=\sum_{k=-L}^{L}w_{k}^{*}\left(t\right)x\left(t-k\right)$$
where, (H) denotes the conjugate transposition, w(t) is the weight vector, and x(t) is the input vector of FIR filter. k∈(−L,L) denotes the filter. The order number of filter is 2*L*+1. Because of the platform moving in the airborne PBR, the multipath signal has the Doppler frequency that is different from the Doppler frequency of the direct-path signal. Thus, the goal of equalization processing is to restore not only the constant modulus but also the original Doppler frequency. The traditional CMA pays attention to keep the signal modulus constant. We consider to modify the blind equalization algorithm by introducing the phase constraint into the cost function. The high-order moment information and the cyclostationarity frequency are used to restrict the phase of output signal. The cost function of modified blind equalization algorithm is shown as: (13)$$\begin{array}{cc} \min & E_{t}\left\{{\left[{\left|\hat{y}\left(t\right)\right|}^{2}-\gamma_{2}\right]}^{2}\right\}+\sum_{k=1}^{4}a_{k}{\left(E_{t}\left\{{\hat{y}}^{k}\left(t\right)\right\}-E_{t}\left\{s^{k}\left(t\right)\right\}\right)}^{2}\\ s.t. & E_{t}\left\{\hat{y}\left(t\right)\cdot \hat{y}\left(t\right)e^{-j2\pi 2f_{d}t}\right\}=1 \end{array}$$
where, $\gamma_{2}=E_{t}\left\{{\left|s\left(t\right)\right|}^{4}\right\}/E_{t}\left\{{\left|s\left(t\right)\right|}^{2}\right\}$ is the modulus information of source signal s(t). In this paper, the source signal s(t) is the direct-path signal. $\hat{y}\left(t\right)$ is the equalized output. $E_{t}\left\{{\left[{\left|\hat{y}\left(t\right)\right|}^{2}-\gamma_{2}\right]}^{2}\right\}$ is the cost function of traditional CMA. *k* denotes the order number of moment, and $a_{k}$ is the positive parameter that provides a tradeoff of high-order moments. $E_{t}\left\{\hat{y}\left(t\right)\cdot \hat{y}\left(t\right)e^{-j2\pi 2f_{d}t}\right\}=1$ is used as the restriction factor to keep the Doppler frequency of equalization output, and $f_{d}$ is the cyclostationarity frequency of direct-path signal.

The Lagrangian function is shown as:(14)$$L\left(w,\lambda \right)=E_{t}\left\{{\left[{\left|\hat{y}\left(t\right)\right|}^{2}-\gamma_{2}\right]}^{2}\right\}+\sum_{k=1}^{4}a_{k}{\left(E_{t}\left\{{\hat{y}}^{k}\left(t\right)\right\}-E_{t}\left\{s^{k}\left(t\right)\right\}\right)}^{2}+\lambda \left(E_{t}\left\{\hat{y}\left(t\right)\cdot \hat{y}\left(t\right)e^{-j2\pi 2f_{d}t}\right\}-1\right)$$
where, the first part of Equation (14) is the cost function of traditional CMA to constrain the output modulus, the second part is the high-order moment information, and the third part is the constraint of Doppler frequency.

To solve the complex constraint optimizing issue, the complex value BP network is exploited to equalize the contaminated reference signal. The structure of BP network is shown as Figure 2. The forward network trained by BP algorithm can be named as BP network. The BP network is composed of the input layer, the hidden layer, and the output layer. In general, the BP network is the multi-layer feedforward neural network trained by BP algorithm. In BP network, the error signal propagates back and the signal propagates forward. BP network utilizes the gradient descent algorithm to update the parameters of network.

In this paper, the network with three layers is applied. The process of data through the BP network is expressed as following. The contaminated reference signal, as the input of BP network, multiplies the weight connecting the input layer with the hidden layer, and the summed product is input to the hidden layer. In the hidden layer, if the input is larger than the threshold, the neuron will output the data by the active function. The output of hidden layer multiplies the weight connecting the hidden layer with the output layer, and the summed product is input to the output layer. If the neuron of output layer is activated, the neuron outputs the equalized reference signal. Then the parameter of network will be updated by the error signal of the cost function. 

The output of *h*th neuron in the hidden layer and the output of neuron in the output layer are shown as:(15)$$\begin{array}{l} b_{h}=f\left(\alpha_{h}-\gamma_{h}\right)\\ \hat{y}=f\left(\beta -\theta \right) \end{array}$$
where, f(⋅) is the activate function of neuron, the sigmoid function is used as the active function in this paper. $\gamma_{h}$ is the threshold of the *h*th neuron in the hidden layer, and θ is the threshold of neuron in the output layer. $\alpha_{h}$ and β are the input of the *h*th neuron in the hidden layer and the neuron in the output layer. $\alpha_{h}$ and β are shown as:(16)$$\begin{array}{l} \alpha_{h}=\sum_{i=1}^{d}v_{ih}x_{i}\\ \beta =\sum_{h=1}^{q}w_{h}b_{h} \end{array}$$
where, $v_{ih}$ is the weight to connect the *i*th neuron in the input layer with the *h*th neuron in the hidden layer, $w_{h}$ is the weight to connect the *h*th neuron in the hidden layer with the output neuron.

According to the proposed modified blind equalization algorithm, the error signal of training sample is shown by Equation (14). The gradient descent algorithm is applied to update the weight of BP network, and the parameters are iterated with the negative gradient. For example, the iterative formula of $w_{h}$ is given by: (17)$$\Delta w_{h}=-\eta \frac{\partial L}{\partial w_{h}}=-\eta \frac{\partial L}{\partial \hat{y}}\cdot \frac{\partial \hat{y}}{\partial \beta }\cdot \frac{\partial \beta }{\partial w_{h}}$$
where, η is the iterative step parameter. The iterative formula can be separated into three parts. The last part of iterative formula is shown as:(18)$$\frac{\partial \beta }{\partial w_{h}}=b_{h}$$

Then, the character of activate function is shown by: $f^{\prime }\left(x\right)=f\left(x\right)\left(1-f\left(x\right)\right)$, so the middle part of iterative formula can be derived as:(19)$$\frac{\partial \hat{y}}{\partial \beta }=f^{\prime }\left(\beta -\theta \right)=f\left(\beta -\theta \right)\left(1-f\left(\beta -\theta \right)\right)=\hat{y}\left(1-\hat{y}\right)$$
Finally, the front part of iterative formula is derived as:(20)$$\frac{\partial L}{\partial \hat{y}\left(t\right)}=4\left[{\left|\hat{y}\left(t\right)\right|}^{2}-\gamma_{2}\right]\hat{y}\left(t\right)+2\sum_{k=1}^{4}\frac{a_{k}}{t}k{\hat{y}}^{k-1}\left(t\right)\left(E_{t}\left\{{\hat{y}}^{k}\left(t\right)\right\}-E\left\{s^{k}\left(t\right)\right\}\right)+2\lambda \hat{y}\left(t\right)e^{-j2\pi 2f_{d}t}$$
where, *k*-order moment can be computed by the iteration equation:(21)$$E_{t}\left\{{\hat{y}}^{k}\left(t\right)\right\}=\frac{1}{t}\left[\left(t-1\right)E_{t}\left\{{\hat{y}}^{k}\left(t-1\right)\right\}+{\hat{y}}^{k}\left(t\right)\right])$$

Let it $g=-\frac{\partial L}{\partial \hat{y}}\cdot \frac{\partial \hat{y}}{\partial \beta }$, and the weight and the threshold can be derived as:(22)$$\begin{array}{l} \Delta w_{h}=\eta gb_{h}\\ \Delta \theta =-\eta g\\ \Delta v_{ih}=\eta e_{h}x_{i}\\ \Delta \gamma_{h}=-\eta e_{h} \end{array}$$
where, $e_{h}$ can be given by:(23)$$\begin{array}{cc} e_{h}= & -\frac{\partial L}{\partial b_{h}}\cdot \frac{\partial b_{h}}{\partial \alpha_{h}}\\ & =-\frac{\partial L}{\partial \beta }\cdot \frac{\partial \beta }{\partial b_{h}}\cdot f^{\prime }\left(\alpha_{h}-\gamma_{h}\right)\\ & =w_{h}\cdot g\cdot f^{\prime }\left(\alpha_{h}-\gamma_{h}\right)\\ & =b_{h}\left(1-b_{h}\right)\cdot w_{h}\cdot g \end{array}$$

The modified blind equalization algorithm based on the cyclostationarity is derived and the feasibility is proved in the next section.

## 4. Simulation

The LFM pulse signal is chosen to prove the feasibility of the proposed algorithm in this paper. According to the known flight information of platform and the position of illuminator, the Doppler frequency of direct-path signal is compensated in this simulation. The uniform linear array (ULA) is exploited to receive the echoes. The neuron numbers of input, hidden, and output layers are set as [1,8,10] respectively. The interval of adjacent array elements is the half-wavelength. The other simulation parameters are set in Table 1.

### 4.1. Comparison of Spatial-Temporal Clutter Spectrum

In this simulation, we evaluate the performance of proposed algorithm in term of the spatial-temporal clutter spectrum. In the comparison experiment, the clutter spectrums in four cases are presented in Figure 3. Case 1 demonstrates the clutter power spectrum with the ideal reference signal, Case 2 demonstrates the clutter power spectrum with the contaminated reference signal caused by the multipath signal, Case 3 demonstrates the clutter power spectrum with the contaminated reference signal equalized by the traditional CMA, and Case 4 demonstrates the clutter power spectrum with the contaminated reference signal equalized by the modified blind equalization algorithm in this paper. The full STAP, AEP-STAP, JDL-STAP, and 3DT-STAP are used to evaluate the clutter spectrum in the comparison experiment.

Compare Case 1 and Case 2, the expanded clutter “ridge” caused by the multipath signal is obvious in the clutter power spectrum. The reason is that the Doppler frequency of multipath signal brings the different spatial-temporal character, so there are the expanded clutter sidelobes on two sides of the “high-light” clutter mainlobe. The expanded clutter spectrum will impact the clutter suppression performance of STAP. As can be seen from Case 3 and Case 4, the expansion level of clutter “ridge” is restricted. The reason is that the multipath signal in the reference signal is suppressed by the blind equalization processing, and the expanded clutter caused by the multipath is weakened. Comparing Case 3 and Case 4, the proposed algorithm in this paper obtains the more perfect spatial-temporal clutter spectrum. The simulation proves the feasibility and superiority of proposed method.

### 4.2. Comparison of Improve Factor (IF)

In this simulation, the improved factor (IF) is used to prove the feasibility of the proposed algorithm. The IF is defined as the ratio of signal-noise ratio (SNR) between output and input of the adaptive filter, and the IF is usually used to measure the performance of STAP. The expression of IF is shown as:(24)$$\mathrm{IF}=\frac{P_{s}^{\mathrm{out}}/P_{n}^{\mathrm{out}}}{P_{s}^{\mathrm{in}}/P_{n}^{\mathrm{in}}}=\frac{w^{*}ss^{*}w/w^{*}Qw}{s^{*}s/\mathrm{tr}\left(Q\right)}=\frac{w^{*}ss^{*}w\cdot \mathrm{tr}\left(Q\right)}{w^{*}Qw\cdot s^{*}s}$$
where, $P_{s}^{\mathrm{out}}/P_{n}^{\mathrm{out}}$ denotes the SNR of adaptive filter output, $P_{s}^{\mathrm{in}}/P_{n}^{\mathrm{in}}$ denotes the SNR of adaptive filter input, w is the weight vector of STAP, Q is the covariance matrix of snapshot data, and **s** is the space-time steering vector.

In this simulation, the IF curves is compared in four scenarios. Figure 4 demonstrates the comparison of IF curves. In Figure 4, “proposed algorithm” denotes the modified blind equalization algorithm in this paper, “without equalization” denotes the contaminated reference signal is used into the signal processing of airborne PBR directly, “traditional algorithm 1” denotes the CMA which has been introduced in Equation (13), and “proposed algorithm 2” denotes the normalized CMA (NCMA). In every iteration step of NCMA, the normalization processing is made for the better convergence, and the derivation process of NCMA can be found in [27].

As can be seen from Figure 4, the IF curve without equalization has the widest clutter suppression region where the targets can be removed. The clutter suppression region partly shrinks in the IF curve of traditional Algorithm 1 and 2, and the proposed algorithm can provide the narrowest suppression region in the IF curve. The reason is analyzed. The traditional algorithm 1 and 2, which only keep the constant modulus of equalized signal, are not suitable for the multipath signal with the Doppler frequency. The proposed algorithm applies the cyclostationarity frequency to limit the Doppler frequency of multipath signal, so the performance of proposed algorithm is better than traditional Algorithm 1 and 2. According to the IF curve of proposed algorithm, the clutter can be suppressed in the concave area and the targets can be detected in the flat area. The simulation proves that the modified equalization algorithm proposed in this paper can improve the detection performance of STAP.

### 4.3. Comparison of Target Detection Performance

This simulation is constructed to prove the availability of proposed algorithm in term of the target detection performance. Two simulated targets are injected into the 700th range bin, and the normalized spatial and Doppler frequency of target 1 and 2 are set as [0, −0.1] and [0.2, 0.35]. The simulation results with four kinds of STAP algorithms are shown in Figure 5. The target areas are marked by the red dotted lines. 

Figure 5a,c,e,g demonstrates the detection outputs with the reference signal contaminated by the multipath signal. As can be seen from Figure 5a,c, the areas marked by red dotted lines are blurry and no target can be detected. The similar simulation result is shown in Figure 5e,g, it is difficult to accomplish the detection of target 1, and the detection performance of target 2 degrades into the lower level. The reason is that the targets are close to the expanded clutter in the spatial-temporal dimension. When the expanded clutter is eliminated by STAP algorithm, the targets are also suppressed. This simulation illustrates that the contaminated reference signal deteriorates the target detection performance severely. Figure 5b,d,f,h shows the detection outputs of the full STAP, AEP-STAP, JDL-STAP, and 3DT-STAP with the contaminated reference signal equalized by the proposed algorithm in this paper. As can be seen from Figure 5b,d,f,h, the peaks of simulated targets can be detected in the marked areas clearly. The reason is that the multipath element in the contaminated reference signal is suppressed by the proposed algorithm, then the expanded clutter spectrum caused by the multipath signal is improved effectively. Thus, the targets are not impacted by STAP. This simulation proves the proposed algorithm can improve the target detection performance in the case of contaminated reference signal.

### 4.4. False Target Removal

This simulation is constructed to prove that the proposed algorithm can remove the false targets caused by the multipath signals. In this simulation, the impact of setting multipath signal 1 is considered and the simulated targets are set as same as Section 4.3. According to the time delay and the Doppler frequency of multipath signal 1, the derived false targets are located in the 670th range bin and the Doppler frequencies of false targets are shifted.

The detection output of 670th range bin is shown as Figure 6. Figure 6a shows the detection output of full STAP with the contaminated reference signal, and Figure 6b shows the detection output with the contaminated reference signal equalized by the proposed algorithm. The false targets are marked by the blue full lines, and the red dotted lines mark the positions of real targets in the 700th range bin. In Figure 6a, the false target 1 that is derived from the real target 1 of 700th range bin emerges in the blue-line frame. Compared with the real target 1, the location of false target 1 shifts to the left in the normalized Doppler frequency, and the shift value is correlated with the Doppler of multipath signal 1. The false target 2 that is derived from the real target 2 is located in the extended clutter region and the false target 2 is suppressed by STAP. In Figure 6b, it is clear there is no high-light peak in the false target position marked by the blue-line frame. As can be seen from the comparison of Figure 6, the derived false targets are eliminated by the proposed algorithm. The comparison result proves that the contaminated reference signal that causes the false targets can be equalized by the proposed algorithm effectively.

The Doppler dimension and the spatial dimension of false target 1 are shown as Figure 7 in detail. The green line and the purple line denote the output without and with the proposed algorithm separately. As can be seen from Figure 7, the peak of false target marked by the green line is obvious in the Doppler dimension and the spatial dimension, and the peak of false target marked by the purple line is lower with the proposed algorithm. This simulation proves that the proposed method can remove the derived false targets caused by the multipath signals effectively.

### 4.5. Algorithm Performance Analysis

In this simulation, the performance of proposed algorithm is analyzed. The multipath number and the clutter noise radio (CNR) of multipath signals in the reference signal are two important factors to be considered. In this simulation, the similarity coefficient is exploited to measure the performance of equalization algorithm proposed in the paper. The expression of similarity coefficient is given by:(25)$$\zeta \left(\hat{y},y\right)=\frac{\left|\sum_{t=1}^{N}\hat{y}\left(t\right)y\left(t\right)\right|}{\sqrt{\sum_{t=1}^{N}{\hat{y}}^{2}\left(t\right)\sum_{t=1}^{N}y^{2}\left(t\right)}}$$
where, $\hat{y}\left(t\right)$ is the equalized signal by the proposed algorithm, and y(t) is the source signals which is the desired direct-path signal. The lager similarity coefficient means the equalized signal is closer to the source signal, and the performance of equalization algorithm is better. In this simulation, the multipath number and the CNR are set as the variable factors and the equalization performance of proposed algorithm is analyzed with the similarity coefficient.

In this simulation, the multipath number varies from 4 to 18, and the CNR varies from 15 dB to 40 dB. The Doppler frequencies are set as the random integer from −100 Hz to 100 Hz, and the time delay bins also are set as the random integer from 20 to 60. The result of 500 Monte Carlo experiments is shown as Table 2.

As can be seen from Table 2, as the multipath CNR increases, the similarity coefficient decreases obviously, and the equalization performance worsens. On the other hand, as the number of multipath increases, the similarity coefficient decreases slightly, so the variety of number multipath is the minor factor in the performance of the proposed algorithm. The conclusion can be drawn that the multipath CNR plays a major role in the performance of the proposed algorithm.

## 5. Conclusions

In the airborne PBR system, the transmitted sample needs to be received as the reference signal by the reference channel toward the opportunity illuminator. The received reference signal is usually impacted by the multipath signal in the actual scenario. Because of the receiver platform moving, the multipath signal in the contaminated reference signal owns the different Doppler frequency. The contaminated reference signal can bring the expanded spatial-temporal clutter spectrum and the false targets. The range bin of false target is related with the time delay of multipath signal. Some algorithms have been proposed to suppress the additional spatial-temporal clutter caused by the multipath signal, such as the expanded clutter suppression algorithm based on sparse representation, but the issue of false target is ignored.

The authors in this paper consider solving the issue of contaminated reference signal by the blind equalization processing, but the existing equalization algorithm is unavailable for the multipath signal with the Doppler frequency. Aiming at the issue of contaminated reference signal in the airborne PBR, the novel blind equalization algorithm is proposed in this paper. The proposed algorithm introduces the high-order moment and the cyclostationarity into the cost function of traditional CMA for restricting the Doppler frequency of multipath signal. The complex BP neural network is exploited to solve the constraint optimization problem for the better convergence. When the multipath elements in the contaminated reference signal are suppressed by the proposed algorithm, the expanded clutter and the false targets can be removed. The impact of contaminated reference signal is eliminated, and the target detection performance is improved. Finally, the simulation experiments prove the feasibility of the proposed algorithm in this paper. However, in this paper, the authors only consider the LFM pulse signal as the opportunity signal, and the case of coding signal will be researched and the parameters of network will be optimized in the following works.

## Figures and Tables

**Figure 1 sensors-20-00788-f001:**
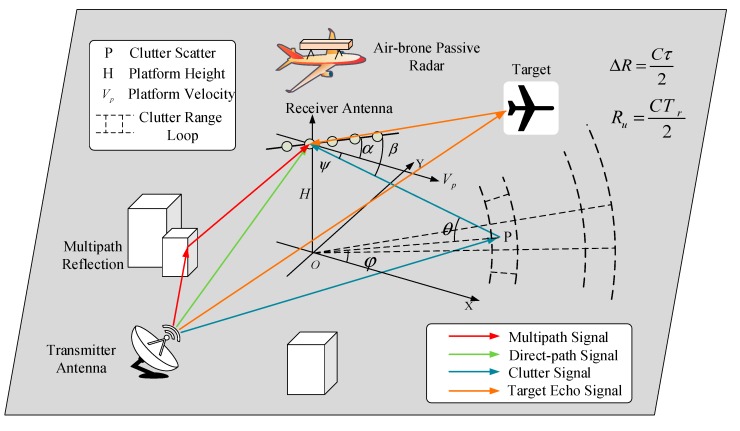
Structure of airborne passive bistatic radar (PBR) system.

**Figure 2 sensors-20-00788-f002:**
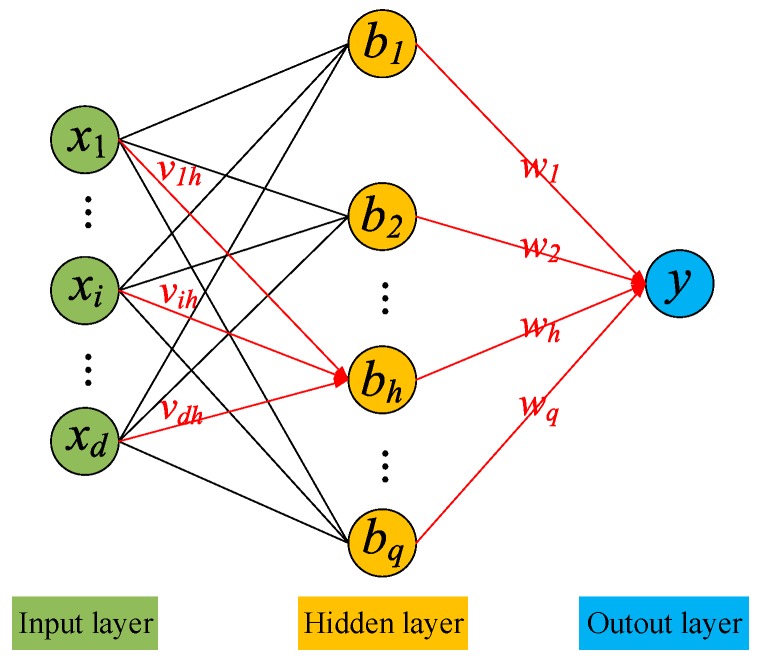
Structure of back propagation (BP) network.

**Figure 3 sensors-20-00788-f003:**
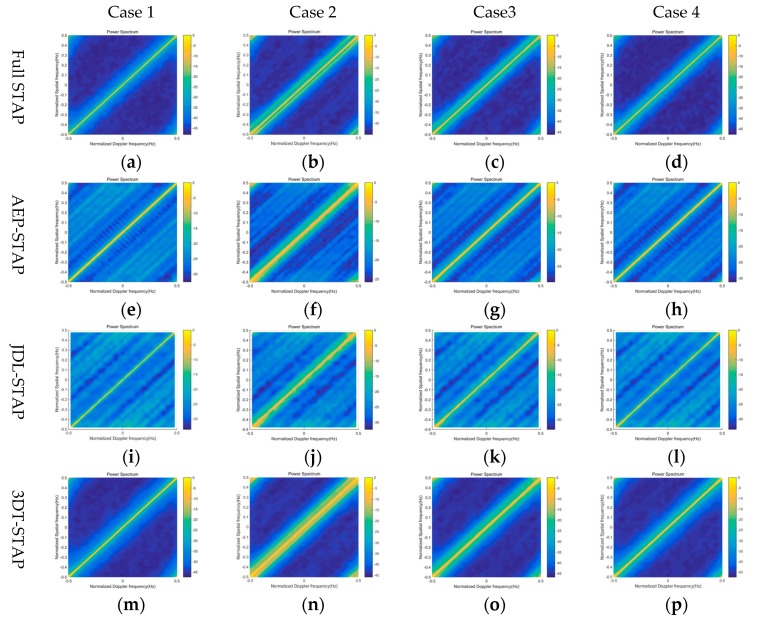
Spatial-temporal clutter spectrum in different cases: (**a**) Full STAP in case 1; (**b**) full STAP in case 2; (**c**) full STAP in case 3; (**d**) full STAP in case 4; (**e**) AEP-STAP in case 1; (**f**) AEP-STAP in case 2; (**g**) AEP-STAP in case 3; (**h**) AEP-STAP in case 4; (**i**) JDL-STAP in case 1; (**j**) JDL-STAP in case 2; (**k**) JDL-STAP in case 3; (**l**) JDL-STAP in case 4; (**m**) 3DT-STAP in case 1; (**n**) 3DT-STAP in case 2; (**o**) 3DT-STAP in case 3; (**p**) 3DT-STAP in case 4.

**Figure 4 sensors-20-00788-f004:**
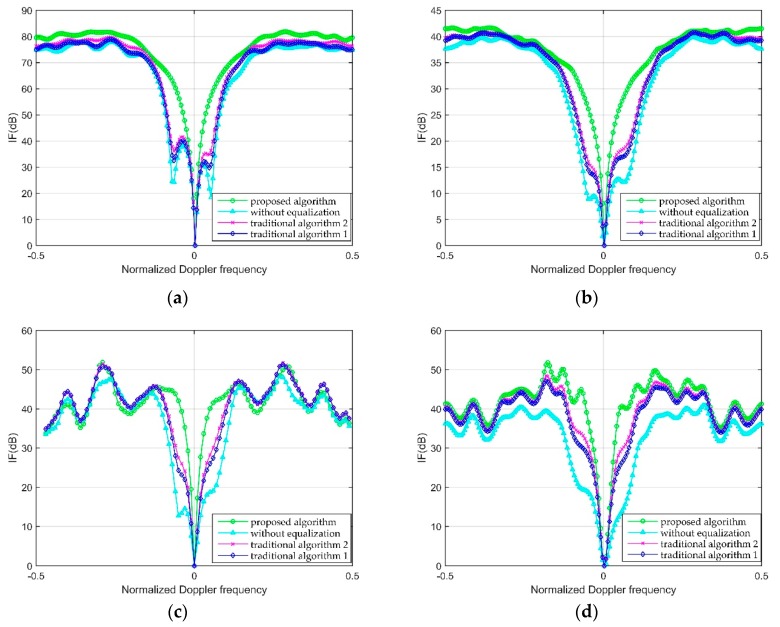
IF comparison with different algorithms: (**a**) Full STAP; (**b**) 3DT-STAP; (**c**) JDL-STAP; (**d**) AEP-STAP.

**Figure 5 sensors-20-00788-f005:**
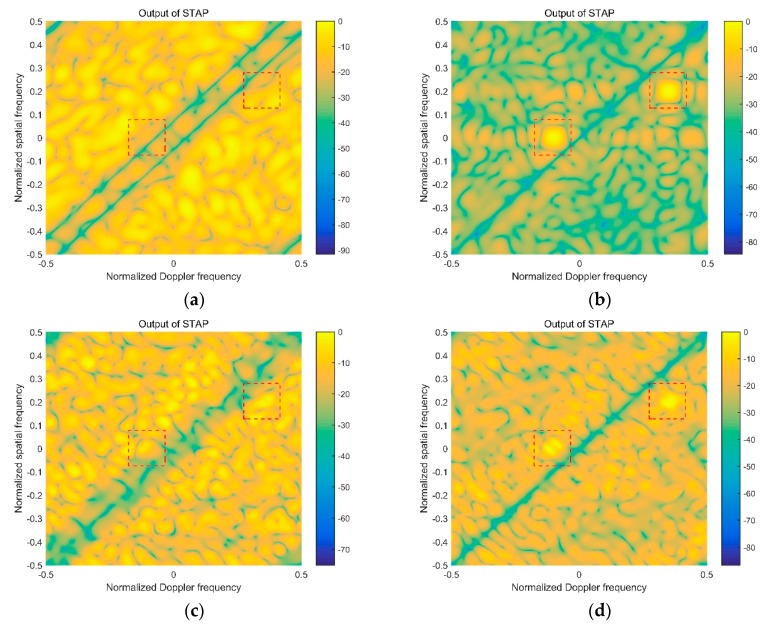
Comparison of target detection performance: (**a**) Full STAP before equalization; (**b**) full STAP after equalization; (**c**) AEP-STAP before equalization; (**d**) AEP-STAP after equalization; (**e**) JDL-STAP before equalization; (**f**) JDL-STAP after equalization; (**g**) 3DT-STAP before equalization; (**h**) 3DT-STAP after equalization.

**Figure 6 sensors-20-00788-f006:**
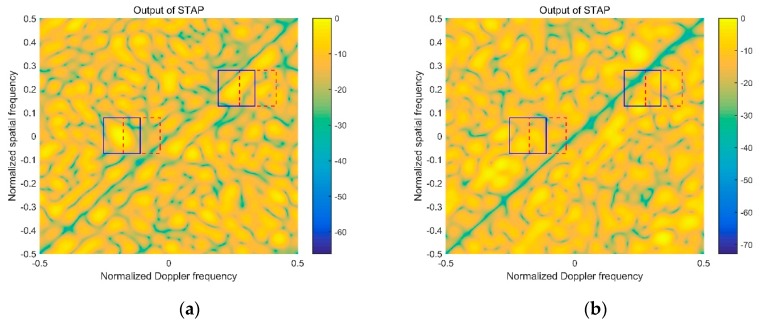
STAP output of 670th range bin: (**a**) Full STAP before equalization; (**b**) full STAP after equalization.

**Figure 7 sensors-20-00788-f007:**
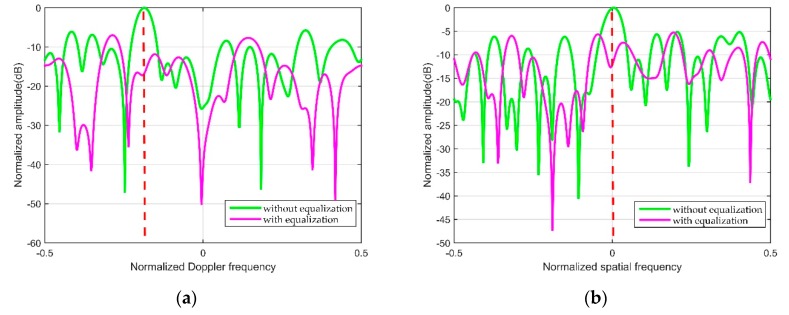
Doppler and spatial dimension of false target: (**a**) Normalized Doppler dimension; (**b**) normalized spatial dimension.

**Table 1 sensors-20-00788-t001:** Simulation parameters.

Parameter	Value
PRF	1000
Bandwidth	2.046 MHz
Wavelength	0.25 m
Platform velocity	125 m/s
Platform height	8000 m
Antenna elements number	20
Pulse number in one CPI	20
Number of multipath	2
Time delay of multipath	[30, 60]
Doppler of multipath	[80 Hz, −50 Hz]

**Table 2 sensors-20-00788-t002:** Similarity coefficient in different cases.

ζ		Number of Multipath							
		4	6	8	10	12	14	16	18
**CNR/dB**	**15**	0.8758	0.8746	0.8655	0.8647	0.8633	0.8630	0.8627	0.8615
	**20**	0.8591	0.8566	0.8554	0.8550	0.8477	0.8451	0.8429	0.8418
	**25**	0.8168	0.8113	0.8111	0.8118	0.8083	0.8078	0.8072	0.8049
	**30**	0.7792	0.7763	0.7734	0.7712	0.7703	0.7696	0.7697	0.7674
	**35**	0.7283	0.7253	0.7267	0.7252	0.7217	0.7174	0.7160	0.7144
	**40**	0.6581	0.6559	0.6518	0.6470	0.6464	0.6433	0.6418	0.6421

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
