# Peer review of "Modified Blind Equalization Algorithm Based on Cyclostationarity for Contaminated Reference Signal in Airborne PBR"

_sensors, 2020, doi:10.3390/s20030788_

Round 1
Reviewer 1 Report
This paper considers interference suppression in an airborne PBR. A modified blind equalization algorithm and complex value back propagation (BP) neural network are used. Some disciplined development and meaningful simulation results are provided. I believe this paper can be accepted after my following concerns are addressed.
- The technical novelties of this work have not been clearly explained. For example, is there anything new with the BP network? Also, it seems unclear to me why the proposed method can achieve an improved performance.
- It is noticed that only one traditional method is compared in the simulation of Section 4. The statistical analysis in Section 4.4 is also insufficient. Additionally, I believe that the effectiveness of the proposed scheme should be tested against real data.
- Many important and relevant literatures are ignored. For instance, for the interferences that owns not only the time delay but also the Doppler frequency in PBR, multi-channel adaptive filters can be adopted [Ref1, Ref2]. Joint spatiotemporal domain filtering can also be helpful [Ref3]. Please cite these references and discuss this issue.
- The language usage of this manuscript should be highly polished.
[Ref1] Sea clutter cancellation for passive radar sensor exploiting multi-channel adaptive filters, IEEE Sensors Journal, 19, 3 (Feb. 2019), 982–995.
[Ref2] A multi-channel partial-update algorithm for sea clutter suppression in passive bistatic radar, 2018 IEEE Sensor Array and Multichannel Signal Processing (SAM) Workshop, pp. 252–256, Sheffield, UK, July 8–11, 2018.
[Ref3] Interference suppression using joint spatio-temporal domain filtering in passive radar, Presented at the IEEE International Radar Conference, pp.1156–1160, Arlington, VA, USA, May 11–15, 2015.
Reviewer 2 Report
Aiming at the Doppler frequency of multipath signal, this paper proposed a new cost function for the phase constraint, and the complex value back propagation (BP) neural network is exploited to solve the constraint optimization problem. I think the idea of this paper is novel. However, before accepting for publication, there are some issues should be addressed.
InEquation (12) and (13), more explains should be given. For example, the parameter of k in Equation (12) belongs to [-L,L], but k in Equation (13) belongs to [1,4], and why the largest value of k is 4. How to explain the constraint factor and the parameter s(t) in Equation (13). In addition, the relationship between Section-3.2 and Section-3.1 should be strengthened.
The sentence “… the multipath signal has the Doppler frequency which is different form the Doppler frequency of direct-path signal”. If there is another moving target, whether the Doppler frequency of multipath signal from this moving target is same as the Doppler frequency of direct-path signal?
When you used neural network, the input and output should be introduced clearly. Besides, there are should be some analysis about the effect of multi-layers and nodes about BP networks on performance. What kind of the activate function of neuron is chosen in your work? How many hidden layers and nodes in your network, and why? At same time, the process of data through the BP network should be explained clearly.
If you can, more other related works should be introduced and analysis in your experiments. In addition, “Traditional Algorithm” should be introduced clearly.
As an important step, I think the process of network should not be ignored, and please should use some words to introduce them.
Reviewer 3 Report
The paper presents an interesting and useful extension in the area of airborne PBR based on the blind equalization. The paper clearly described the goals of the research, theoretical parts of the research. The description of the theoretical part is easy to understand.
I need to highlight the usage of back-propagation neural network for high demanded blind equalization algorithm based on the cyclostationarity algorithm.
The simulation part should be extended for more scenarios to the validation of generality of the algorithm.
Discussion part of the paper is missing.
Round 2
Reviewer 1 Report
I feel that the authors have adequately addressed all my concerns. I thus recommend accept as is.
Reviewer 3 Report
The extension of simulation results is sufficient.